# A cluster randomised controlled trial of community groups using Participatory Learning and Action to prevent and control diabetes and intermediate hyperglycaemia in rural Bangladesh

Edward Fottrell[1]*, Abdul Kuddus[2], Joanna Morrison[1], Tasmin Nahar[2], Carina King[3], Sanjit Kumer Shaha[2], Malini Pires[1], Sarker Ashraf Uddin Ahmed[2], James Beard[4], Naveed Ahmed[2], Andrew Copas[1], Hassan Haghparast-Bidgoli[1], A.K. Azad Khan[2], Kishwar Azad[2]

1 University College London Institute for Global Health, London, United Kingdom, 2 Centre for Health Research and Implementation, Diabetic Association of Bangladesh, Dhaka, Bangladesh, 3 Department of Global Public Health, Karolinska Institutet, Stockholm, Sweden, 4 Independent Consultant, Guildford, United Kingdom

* e.fottrell@ucl.ac.uk

## Abstract

Community mobilisation through Participatory Learning and Action (PLA) has been shown to be effective for a range of health outcomes, including diabetes. Using a cluster randomised controlled trial we evaluated the impact of a PLA community mobilisation intervention for diabetes prevention and control when implemented in rural Bangladesh in 2021–2022 and adapted to the evolving context of the COVID-19 pandemic. 108 PLA groups held a minimum of 13 meetings over a total implementation period of 20 months. Random cross-sectional samples of adults (aged ≥30 years) at baseline (pre-intervention, n = 1392) and at endline (post-intervention, n = 1343) were selected to evaluate intervention impact on the primary outcome of prevalence of intermediate hyperglycaemia and diabetes assessed through fasting blood glucose concentrations and two-hour oral glucose tolerance tests. Secondary outcomes included blood pressure, knowledge of diabetes, and behavioural outcomes, including diet, physical activity, and care-seeking. Results showed no evidence of an intervention effect on prevalence of intermediate hyperglycaemia and diabetes between study arms (adjusted difference -1.19% (95% CI -10.74, 8.36), p = 0.784). Large increases in diabetes knowledge were recorded and the intervention was associated with a significant reduction in mean diastolic blood pressure (-2.98 (-5.55, -0.41), p = 0.028) and possible reductions in the prevalence of hypertension (-7.40% (-16.03, 1.24), p = 0.0845) and abdominal obesity (-15.23% (-33.02, 2.56), p = 0.085). Although COVID-related interruptions and adaptations to implementation of PLA may have impacted effectiveness, the PLA approach to non-communicable disease prevention and control continues to show promise in resource-poor settings.

**Data availability statement:** Deidentified data collected for this study and a data dictionary are available from the UCL Research Data Repository DOI:https://doi.org/10.5522/04/28751210.

**Funding:** This work was funded by the Medical Research Council UK (MR/T023562/1 to EF) under the Global Alliance for Chronic Diseases Scale-Up Programme. The funders had no role in study design, data collection and analysis, decision to publish, or preparation of the manuscript.

**Competing interests:** The authors have declared that no competing interests exist.

Trial registration: ISRCTN42219712 – registered 31st October 2019. Status: complete.

## Introduction

Globally, an estimated 537 million adults aged 20–79 have diabetes and more than three quarters of these people live in low- and middle-income countries [1–3]. By 2045, the International Diabetes Federation predicts that 1 in 8 adults in this age range – approximately 783 million – will be living with diabetes, an increase of 46% and with the largest increases occurring in resource-poor settings [3]. The majority of people living with diabetes in low- and middle-income settings are undiagnosed and unaware of their condition, and health systems in resource-poor settings are ill-equipped to respond to the growing burden of diabetes risk and morbidity [4]. There is therefore a pressing need for effective, scalable and sustainable population-level initiatives to raise awareness and to prevent and control diabetes globally, and particularly in resource-poor settings.

In Bangladesh, the prevalence of diabetes is estimated to range from 2% to 13% and the estimated prevalence of intermediate hyperglycaemia (impaired fasting glucose or impaired glucose tolerance indicating an increased risk of diabetes) varies between 2% and 22% [5]. Between 2015 and 2018, the DMagic (Diabetes Mellitus Action through community Groups or mHealth Information for better Control) trial used a Participatory Learning and Action (PLA) approach for type-2 diabetes mellitus (T2DM) prevention and control directed at the general population in rural Bangladesh [6]. After 18 months of implementation of 122 monthly PLA meetings, the trial showed significant increases in diabetes knowledge and awareness and large reductions in the prevalence of diabetes and intermediate hyperglycaemia, with an absolute reduction of -20.1% (95% CI:-26.1, -14.0) [6,7]. Among individuals identified with intermediate hyperglycaemia, the two-year cumulative incidence odds of diabetes was 59% lower (OR: 0.41; 95% CI:0.24, 0.67), equating to an absolute reduction of -8.4% (95% CI: -13.8, -2.99) [6]. Impact was shown to be highly cost effective, equitable and [8], though diabetes outcomes were no longer observed at 5-years post randomisation, other longer-term potential health impacts were notable, including reductions in population hypertension outcomes [9].

Given the need for context-appropriate interventions to prevent diabetes at scale, the Diabetes: Community-led Awareness, Response and Evaluation (D:Clare) trial assesses the impact of PLA on T2DM and other non-communicable disease and risk outcomes when implemented across a sub-district ('upazila') in Bangladesh and in the context of the evolving COVID-19 pandemic, with pragmatic adaptations to intervention delivery and evaluation [10,11].

## Methods

### Ethics statement

The D:Clare trial received ethical approval from University College London (4199/007) and the Diabetic Association of Bangladesh (BADAS-ERC/E/19/00276). All survey participants indicated informed consent through signature or thumbprint.

### Setting

The D:Clare trial took place between January 2020 and December 2022 in Alfadanga upazila, Faridpur district, Bangladesh. Alfadanga was purposefully selected as the trial location as it had not been exposed to a PLA intervention previously and is close to an existing field office. Alfadanga has an estimated population of 120,000 people, divided into six administrative unions and has a predominantly agricultural economy [12].

### Study design and participants

Originally designed as a stepped-wedge cluster randomised controlled trial [11] we changed our design to a parallel-arm two group cluster RCT with 1:1 allocation [10] due to COVID-19-related project interruptions and potential impacts on health and contextual factors [13].

Each of the six *unions* in Alfadanga were split in two, forming 12 clusters (population approx. = 10,000 each). Evaluation was based on data collected at baseline (post randomisation but pre-intervention delivery) and endline (post intervention) cross-sectional surveys. To minimise inter-cluster contamination, we employed a 'fried-egg' design for our evaluation surveys, with participants residing in border areas excluded from surveys. Everyone in intervention clusters was eligible to participate in the intervention but evaluation was restricted to non-pregnant female and male permanent residents aged 30 years and above.

### Randomisation and masking

Randomisation took place at a project orientation meeting in Faridpur town attended by community representatives and independent observers. Twelve folded pieces of paper, each with a name one of the clusters, were placed into a container and then drawn by meeting participants. The first six clusters were allocated to intervention, and the subsequent six were allocated to control.

### Intervention

**Participatory Learning and Action.** The intervention was community mobilisation through PLA, which works through facilitated community groups actively engaging communities in identifying the causes of health problems, and working together to design and implement ways to address these health problems, and reflect on their progress [14].

One hundred and eight PLA groups, 9 men's and 9 women's in each intervention cluster - approximately 1 per 200 adults aged 30 years and above – were created with the intention to work through four phases of PLA: 1) problem identification where participants identify and prioritise causes of T2DM and T2DM risk in their community; 2) planning together where groups and communities collectively design strategies to address the causes of T2DM that can be implemented by communities; 3) strategy implementation; 4) participatory evaluation of the strategies which they have implemented. The four-phase cycle was planned to take 18 months, with groups meeting once a month. Due to COVID-related interruptions, however, implementation was modified to allow a minimum of 13 meetings held over a total implementation period of 20 months. Additional COVID-related modifications to the intervention delivery are summarised in S1 Table. Those attending groups were encouraged to share learning and key messages with other members of the community who were unable to attend.

PLA groups were led by 12 facilitators (6 males, 6 females) – two in each cluster – recruited from the intervention areas. They received training on group facilitation and basic health messages related to non-communicable disease prevention and control, with particular focus on T2DM. Facilitators were equipped with a picture card, flip chart and PLA manual with topic-specific information and suggested group activities. The intention was for groups to cover one topic in each meeting, but due to COVID-19 adaptations, multiple topics needed to be covered in single meetings. Topics included information on T2DM, its causes, symptoms, prevention, and control. The PLA manual was adapted from the DMagic trial,

aligned with standard recommendations for awareness, prevention and control of T2DM and updated to incorporate information about preventing COVID-19. Male groups were led by a male facilitator and female groups were led by a female facilitator. Facilitators were salaried and each was expected to coordinate and facilitate an average of 18 group meetings per month. Facilitators were mentored and supported by two participatory group coordinators and, a District Manager based locally, and a Senior Group Intervention Manager based in Dhaka.

**Piloting & interruption.** We conducted a 1-month pilot in intervention areas in December 2020. Experiences from the pilot were shared with study Community Advisory Groups – groups based in each of the study upazilas and comprised of community stakeholders – who were asked to provide input on the acceptability of implementing the PLA intervention after COVID-19 lockdown. Details of this community consultation are published elsewhere and describe key adaptations to the intervention, including restrictions on group size and the implementation of COVID-19 screening and hygiene measures [13]. Routine intervention implementation began in January 2021. A further government COVID-19 lockdown was imposed between April 2021 and August 2021, during which time all field activities, including intervention implementation were paused. Intervention implementation resumed in November 2021 and then continued uninterrupted until September 2022.

**Control.** The 6 control clusters received usual care, which in this context is little or no preventative public health campaigning and care seeking in government or private facilities, often associated with out-of-pocket payment for diagnostics, consultations, and treatments. Commitment to deliver the intervention to all clusters was made at the public orientation meeting (reflecting the original stepped-wedge design). Control clusters started to receive the intervention after the trial evaluation endline survey in November 2022.

### Procedures

**Sampling for evaluation.** We used a three-step sampling approach. Prior to randomisation, we purposefully selected two to six evaluation villages from a central area in each cluster, with the aim to have between 800–1000 households present, and assuming an average of two people aged ≥30 years in each household. Eligible villages were those which do not sit on a border with a neighbouring study cluster, do not act as a major trading centre or administrative centre, and had a minimum of 50 households. The list of villages, and their estimated population sizes was derived from the 2011 Bangladesh census, and administrative maps.

A sampling frame of all the households with at least one non-pregnant permanent resident aged 25 years or above within the selected evaluation villages was generated by household census completed in the evaluation villages in January 2020. Each member of the household aged 30 years or older was listed and a sample of households with at least one adult aged ≥30 years was selected using simple random sampling. Next, one adult aged ≥30 years was selected for inclusion in the survey, again using simple random sampling. New samples of households and individuals were generated from the same sampling frame (same villages) for the baseline survey (4th January to 25th February 2021) and endline survey (5th September to 15th November 2022). This general population sample was used to evaluate population-wide intervention impacts.

In addition, individuals aged 30 years or older identified with intermediate hyperglycaemia in the 2021 baseline survey were followed-up at endline (i.e., an intermediate hyperglycaemia cohort) and this sample was used to evaluate intervention impacts on diabetes incidence in a high-risk population. Therefore, we describe the total endline collection as including a cross-sectional random population sample, and a cohort sample of those with intermediate hyperglycaemia at baseline. Some individuals with baseline intermediate hyperglycaemia were also randomly sampled in the cross-sectional survey at endline – these individuals were considered to belong to both samples.

**Survey procedures.** All sampled individuals in a single village were informed of the requirements of the study and were requested to prepare for data collection in the morning of a specified day following an overnight fast. Data collection took place at respondents' homes or at a convenient location in their village.

Data were collected by six teams of locally recruited fieldworkers comprised of one male and one female with at least secondary education. They underwent 10 days training on survey methods and how to take physical measurements, followed by one-week supervised field practice and piloting in non-study villages in Faridpur district. Data collectors were supervised by two field supervisors with experience in survey methods. Questionnaire data were gathered in Bangla on Android tablets using ODK Collect.

**Individual questionnaire.** Fieldworkers used a structured survey instrument adapted from the WHO Stepwise tool [15] and 2014 Bangladesh Demographic and Health Survey [12] to measure the background demographic and socio-economic characteristics, lifestyle and behavioural risk factors, T2DM awareness indicators and health seeking behaviour and costs of care seeking among study participants. Given the established links between diabetes, depression [16] and anxiety, we used the Patient Health Questionnaire-9 (PHQ-9) and General Anxiety Disorder-7 (GAD-7) mental health screening tools to measure depression and anxiety, respectively [17,18]. Both PHQ-9 and GAD-7 have been applied previously in Bangladesh and similar populations [19,20].

**Physical measurements.** Fieldworkers measured blood glucose using the One Touch Verio Flex Glucometer (Lifescan, Inc., Milpitas, USA) in whole blood obtained by finger prick from capillaries in the middle or ring finger. Individuals with a self-reported prior medical diagnosis of T2DM were invited to provide random blood glucose readings whereas all other participants were asked to provide readings after an overnight fast. All individuals without a prior diagnosis of T2DM then received a 75g glucose load dissolved in 250 ml of water and had a repeat capillary blood test within 5 minutes of 120 minutes post ingestion to determine glucose tolerance status and differentiate between individuals with intermediate hyperglycaemia and those with T2DM according to WHO criteria (S2 Table).

Blood pressure was measured using the OMRON HBP 1100 Professional Blood Pressure Monitor (Kyoto, Japan). Two measurements were taken at approximately 5-minute intervals and the respondent's blood pressure obtained by averaging these measurements. Measurements of height, weight, and waist and hip girth were taken with light clothes without shoes and using standard methods [21]. Survey participants were provided with their blood glucose and blood pressure results and were signposted to health services for further testing, advice and treatment.

**Data management & quality control.** Data quality control was ensured through ODK form design, supervisor observation, and data checking in Dhaka, after which data were uploaded to a cloud server where project data leads (King/Beard) ran Stata scripts for further data verification.

The same procedures were applied at the baseline and endline surveys. Data collectors, supervisors and managers were unaware of randomisation assignments at baseline but field staff may have been able to deduce assignment during data collection at endline. Access to endline data was restricted until collection was complete, at which point the data were available for blinded analysis by the PI (EF) and trial statistician (AC). Data collection, management and analytical procedures were monitored by an independent Data Monitoring Committee (DMC). Trial management was also reviewed by an independent Trial Steering Committee (TSC).

**Process evaluation.** Process evaluation data has been published in detail elsewhere [22]. Briefly, we conducted a mixed methods process evaluation following UK Medical Research Council guidelines to describe the context of intervention and control areas, implementation of the intervention, and mechanisms of how the intervention might have affected health outcomes. In intervention areas, we used project monitoring documents, group meeting minutes, observations, interviews and focus group discussions with the intervention team and group members [23]. Qualitative interviews and discussions were conducted by a process evaluation officer, transcribed and translated to English and analysed using the framework approach [24].

## Outcomes

Our primary outcome was the combined prevalence of intermediate hyperglycaemia (i.e., impaired fasting glucose or impaired glucose tolerance) and T2DM among adults aged 30 years or older. We used WHO definitions and blood

glucose cut-offs for normoglycaemia, impaired fasting glucose, impaired glucose tolerance and T2DM as summarised in S2 Table. Individuals with a self-reported prior diagnosis of T2DM were classified as living with diabetes in our analysis, irrespective of blood glucose readings. The primary outcome is defined only for those in the cross-sectional sample at endline.

Secondary outcomes are defined in S3 Table and relate to self-awareness of diabetic status, knowledge of T2DM, utilisation of diabetic services, depression, anxiety and common non-communicable disease risk factors. We also calculated the cumulative incidence of T2DM among the cohort of individuals with intermediate hyperglycaemia.

We also included several pre-specified exploratory and explanatory outcome measures designed to identify potential mechanisms of intervention effect on population behaviours, health and wellbeing. Among individuals with a self-reported prior diagnosis of T2DM, these included: diabetes control (random blood glucose ≤11.1 mmol/L), self-reported diabetes-related health complications, at least monthly blood glucose testing, perceptions of social stigma, measures of family support or abuse related to diabetes diagnosis, and psychological distress and ability to self-manage diabetes assessed using the Appraisal of Diabetes Scale (ADS) [25]. Among all participants, we measured self-rated health (score 0–100 on a visual analogue scale), tobacco use, betel nut use, sedentary time, time engaged in brisk walking within the previous 7 days, and perceptions of social norms related to dietary practices and female physical activity.

## Sample size and statistical analysis

Due to COVID-19 adaptations, we updated our sample size calculation in 2022 using methods described by Copas and Hooper (2020) [26]. With an average of 116 participants per cluster in the baseline survey (n = 1392), we estimated that 125 participants per cluster in the endline survey would give 78% power to detect a 30% reduction in the primary outcome of combined prevalence of T2DM and intermediate hyperglycaemia, assuming a baseline primary outcome prevalence of 40%, an intra-cluster correlation of 0.02 and an estimated autocorrelation between baseline and endline of 0.4. We therefore sampled 132 respondents for the endline survey, to account for non-participation (n = 1584). Our goal was to also collect data from a cohort of participants with intermediate hyperglycaemia at baseline, giving an endline target sample of 1868.

The main analysis of the primary outcome includes all individuals who provided blood glucose measures at endline from the cross-sectional survey. For the cumulative incidence secondary outcome, analysis includes all individuals in the cohort for whom a baseline blood glucose measurement of intermediate hyperglycaemia was taken and an endline blood glucose measurement was taken. We did not impute any values for missing data since key predictors we might use for imputation (e.g., age, sex, baseline hyperglycaemia) were already included in model adjustments.

The primary analyses for each outcome were undertaken on an intention-to-treat basis. For each outcome, unless otherwise specified, we conducted an 'unadjusted' and 'adjusted' analysis. All analyses included baseline data alongside the endline data to gain precision and account for any baseline imbalance. The 'unadjusted' analysis included adjustment only for the baseline prevalence of the outcome at the cluster level. 'Adjusted' analysis included further adjustment for age as a linear and quadratic term, gender, and baseline hyperglycaemia, as specified *a priori* as likely predictors of intermediate hyperglycaemia and T2DM.

We applied a two-stage cluster summary approach for all outcomes because of the modest number of clusters [27]. In the first stage of the unadjusted analysis a cluster summary of the outcome was calculated for each cluster at baseline and endline. In the second stage the endline cluster summary values were analysed using linear regression including trial arm and baseline cluster summary value as predictors. The coefficient for trial arm represents an intervention effect on the difference scale.

For the adjusted analysis, we first calculated adjusted cluster summary values at baseline and endline by, separately for each time point, fitting a model to predict the outcome including as predictors age, gender and (for endline model only) baseline survey participation, but excluding trial arm and ignoring the clustering in the data. The outcome was then

predicted for all participants (as a probability for binary outcomes) and the cluster mean of the predicted values was calculated for all clusters. Next a residual term was calculated for each cluster – the observed cluster mean minus the mean predicted value. In the second stage the residuals at endline were modelled using linear regression including trial arm and baseline residual as predictors. The coefficient for trial arm represents the adjusted intervention effect on the difference scale. We consider the adjusted analysis primary for all outcomes.

Primary analysis of the primary outcome was conducted blind and independently by the trial PI (EF) and trial statistician (AC) who reported these results to the DMC and Chair of the TSC, after which the identities of the trial arms were revealed, and analysis continued un-blinded.

Pre-specified exploratory analyses of possible interactions between intervention and the following characteristics were undertaken using individual level analysis of endline data: (1) gender, (2) household wealth, (3) age, (4) village size (less than mean village size vs. greater than or equal to mean village size), and (5) inclusion in baseline survey. Specifically, we used independence estimating equations with robust standard errors through the use of the complex survey 'svy' functions in Stata and added the interaction term between allocated group and subgroup variable into a logistic regression (odds) model.

To assess effects of possible misclassification in the primary outcome based on self-reported diagnoses, we also conducted sensitivity analysis of intervention effect on the primary outcome whereby a) all cases of self-reported T2DM were dropped from the analysis and b) all cases were defined based on blood glucose measures only (i.e., disregarding self-reported diagnoses) where individuals with self-reported diabetes with random blood glucose values of ≥11.1 mmol/L were classified as living with diabetes and those with values <11.1 mmol/L were classified as normoglycaemic.

To assess effects of possible blood glucose measurement bias we assessed intervention impact on continuous blood glucose measurements and by separately applying different arbitrary fasting blood glucose cut-offs of 5·5 mmol/L, 6·3 mmol/L, and 7·8 mmol/L and 2-h blood glucose cut-offs of 6·8 mmol/L and 10·4 mmol/L for classifications of intermediate hyperglycaemia or T2DM.

Total cost and cost-effectiveness analysis of the PLA intervention was conducted from a programme provider perspective. Costs were collected prospectively from project accounts and input into a customised excel-based costing tool designed for this purpose. Time horizon for costing was 36 months, including 16 months start-up and 20 months implementation. Costs were estimated. The incremental cost-effectiveness ratio (ICER) was calculated in terms of cost per 1-mmHg reduction in diastolic blood pressure (statistically significant secondary outcome). All costs were adjusted for inflation, discounted at 3% per year and converted to 2022 US dollar (US$) using the exchange rate of 1US$ = 91.75 Bangladeshi Taka [28]. A detailed analysis will be presented in a separate publication.

Statistical analysis was conducted in StataSE version 15 and StataMP version 18, and a 5% significance level used for all analyses.

### Patient & public involvement

Members of the public and community representatives were engaged in a community orientation meeting at the start of the study and in regular community advisory groups to inform the processes and methods used in the study. This included representatives from each of the study upazillas acting as a link between the project team and communities and providing direct input into the design of study tools and methods, intervention delivery (including interruptions and re-starting PLA groups in the context of COVID-19 [13]), and, at the end of the trial, in dissemination and interpretation of study findings.

### Results

Baseline data were collected across all 12 study clusters from 1392/1584 (87.9%) individuals, of whom 326 individuals were identified with intermediate hyperglycaemia. Endline data were collected from 1343/1584 (84.8%) randomly sampled individuals, and 272/326 (83.4%) of the individuals with intermediate hyperglycaemia at baseline (Fig 1).

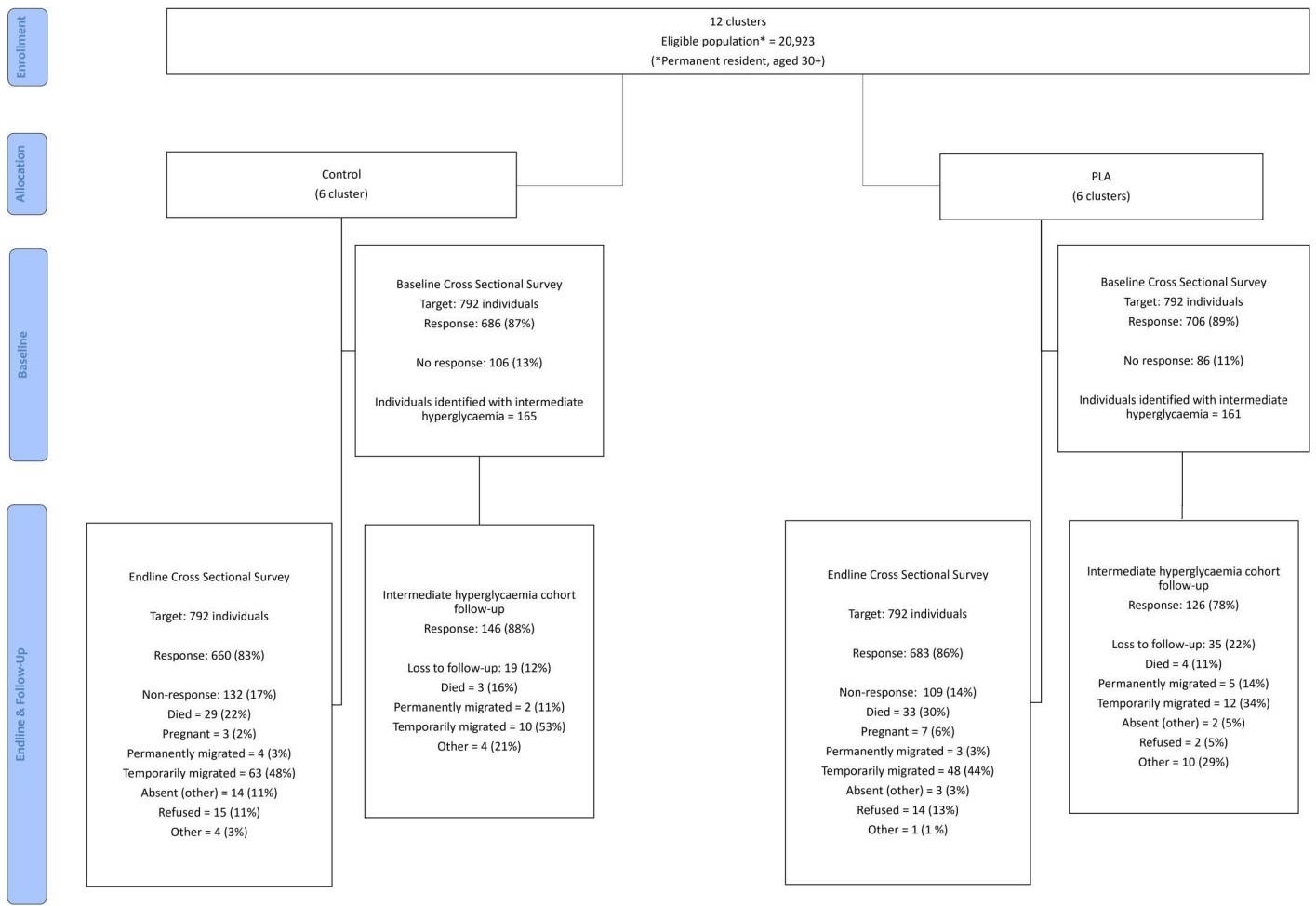

**Fig 1. Trial consort profile.**

Sociodemographic characteristics and blood glucose classifications in the baseline and endline random survey samples by study arm are shown in Table 1.

Responders at baseline and endline were more likely to be female (baseline: 92.7% female vs 80.7% male, p < 0.001; endline: 89.3% female vs. 78.7% male, p < 0.001) than non-responders. Male responders were older than male non-responders (baseline mean difference 4.5 years, p = 0.0026; endline mean difference 1.0 year, p = 0.1333) and female responders were younger than female non-responders (baseline mean difference = 2.4 years, p = 0.1464; endline mean difference 2.2 years, p = 0.0009). The same patterns of non-response were observed across trial arms.

A total of 108 groups were held by 12 facilitators, with each facilitator responsible for eight to eleven groups per month. Each group had an average of 26 participants and based on our COVID protocol, groups were divided into two separate meetings of around 13 participants in each. Around 60% of the attendees were between 30–40 years old, and the remaining were over 50 years of age. COVID-19 disruptions prevented fidelity to the PLA intervention approach and limited the duration of the intervention. Individual and household actions to prevent diabetes were reported, and group members indicated that they had learned enough information over 10 months to implement individual change. However, PLA group attendance was low after the wider community meeting and data suggest that restricting the number of people attending

**Table 1.  Count and cluster mean sociodemographic characteristics and blood glucose classification by trial arm at baseline and endline random population samples.**

| | | Baseline | | Endline | |
|---|---|---|---|---|---|
| | | Control | Intervention | Control | Intervention |
| Clusters | | 6 | 6 | 6 | 6 |
| Average cluster population aged ≥30 years (sd) | | 1750.7 (330.8) | 1782.5 (166.7) | 1818.7 (342.2) | 1840.3 (169.2) |
| Average number of households with eligible members per cluster (sd) | | 975.7 (76.9) | 971.0 (151.8) | 990.3 (76.9) | 991.4 (153.9) |
| Total (survey) | | 686 | 706 | 660 | 683 |
| Age | 30–39 years | 198 (28.9%) | 216 (30.6%) | 116 (17.6%) | 129 (18.9%) |
| | 40–49 years | 183 (26.7%) | 169 (23.9%) | 196 (29.7%) | 163 (23.9%) |
| | 50–59 years | 123 (17.9%) | 121 (17.1%) | 161 (24.4%) | 162 (23.7%) |
| | 60–69 years | 118 (17.2%) | 125 (17.7%) | 90 (13.6%) | 119 (17.4%) |
| | 70–100 years | 64 (9.3%) | 75 (10.6%) | 97 (14.7%) | 110 (16.1%) |
| Sex | Male | 248 (36.2%) | 268 (38.0%) | 243 (36.8%) | 264 (38.6%) |
| | Female | 438 (63.9%) | 438 (62.0%) | 417 (63.2%) | 419 (61.4%) |
| Education | None | 369 (53.8%) | 457 (64.7%) | 364 (55.2%) | 387 (56.7%) |
| | Primary | 231 (33.7%) | 192 (27.2%) | 235 (35.6%) | 231 (33.8%) |
| | Secondary | 71 (10.4%) | 44 (6.2%) | 47 (7.1%) | 48 (7.0%) |
| | Tertiary | 15 (2.2%) | 13 (1.8%) | 14 (2.1%) | 17 (2.5%) |
| Illiterate | Literate | 385 (56.1%) | 354 (50.1%) | 359 (54.4%) | 408 (59.7%) |
| | Illiterate | 301 (43.9%) | 352 (49.9%) | 301 (45.6%) | 275 (40.3%) |
| Marital status | Married | 586 (85.4%) | 596 (84.4%) | 554 (83.9%) | 545 (79.8%) |
| | Not married | 100 (14.6%) | 110 (15.6%) | 106 (16.1%) | 138 (20.2%) |
| Religion | Muslim | 660 (96.2%) | 688 (97.5%) | 614 (93.0%) | 670 (98.1%) |
| | Other | 26 (3.8%) | 18 (2.6%) | 46 (7.0%) | 13 (1.9%) |
| Occupation | Not working | 468 (68.2%) | 457 (64.7%) | 448 (67.9%) | 428 (62.7%) |
| | Manual labour | 183 (26.7%) | 224 (31.7%) | 187 (28.3%) | 229 (33.5%) |
| | Non-manual labour | 35 (5.1%) | 25 (3.5%) | 25 (3.8%) | 26 (3.8%) |
| Wealth quintile | Most poor | 109 (15.9%) | 170 (24.1%) | 131 (19.9%) | 141 (20.6%) |
| | Very poor | 148 (21.6%) | 148 (21.0%) | 145 (22.0%) | 125 (18.3%) |
| | Poor | 143 (20.9%) | 118 (16.7%) | 153 (21.2%) | 121 (17.7%) |
| | Less poor | 167 (24.3%) | 121 (17.1%) | 119 (18.0%) | 142 (20.8%) |
| | Least poor | 119 (17.4%) | 149 (21.1%) | 112 (17.0%) | 154 (22.6%) |
| Total (diabetes outcomes) | | 681 | 704 | 660 | 683 |
| Diabetes outcomes | Normal | 374 (54.9%) | 435 (61.8%) | 331 (50.2%) | 372 (54.5%) |
| | IFG* | 30 (4.4%) | 34 (4.8%) | 31 (4.7%) | 45 (6.6%) |
| | IGT* | 135 (19.8%) | 127 (18.0%) | 154 (23.3%) | 160 (23.4%) |
| | Diabetes | 142 (20.9%) | 108 (15.3%) | 144 (21.8%) | 106 (15.5%) |

*IFG = Impaired Fasting Glucose; IGT = Impaired Glucose Tolerance.

groups, masking, social distancing, and the discomfort around household visits in a COVID-19 pandemic context did not engender community or group cohesion necessary to trigger mechanisms of community-wide actions. Table 2 shows that 31.6% of the intervention arm sample reported ever participating in a PLA group and those who did participate reported attending a median of 2 out of a maximum of 13 meetings. Approximately 1 in 4 respondents who did not attend PLA groups reported knowing someone who did and 16% of respondents participated in the wider community meeting.

**Table 2. PLA intervention coverage process indicators by trial arm.**

| Intervention exposure | Allocation | |
|---|---|---|
| | Control | Intervention |
| Ever participated in PLA group (%) | 0 (0%) | 216 (31.6%) |
| Median number of meetings attended (if>0), (IQR; min; max) | NA | 2 (2–5; min. = 1, max. = 13) |
| Respondent knows someone who attended PLA group if respondent did not attend (%) | 0 (0%) | 115 (24.6%) |
| Participated in community meeting (%) | 1 (0.2%) | 109 (16.0%) |

There was no evidence of an intervention effect on the prevalence of intermediate hyperglycaemia and T2DM (adjusted difference, intervention minus control -1.19 (-10.74, 8.36), p = 0.7838) (Table 3). Findings were similar in pre-specified sensitivity analyses (S4 Table).

Amongst our secondary outcomes, large increases in ability to report one or more valid causes, symptoms, complications and strategies for prevention and control of T2DM were observed in the intervention arm relative to control (Table 3). PLA intervention was associated with a significant reduction in diastolic blood pressure (-2.98 (-5.55, -0.41), p = 0.0277), but little effect on systolic blood pressure, and possible reductions in the prevalence of hypertension (-7.40 (-16.03, 1.24), p = 0.0845) and abdominal obesity (-15.23 (-33.02, 2.56), p = 0.0847). Two-year cumulative incidence of T2DM among the intermediate hyperglycaemia cohort was lower in intervention clusters relative to control, but the difference was not statistically significant (-8.60 (-20.09, 2.89), p = 0.1265). A potential small negative intervention impact on physical activity is noted.

There was a significant increase in self-rated health in the PLA arm relative to control and a large but non-significant reduction the prevalence of self-reported diabetes-related health complications (Table 4). A small potential negative intervention effect on sedentary time was observed, as was a significant increase in perceived negative social or family reactions to eating less than usual or refusing oily or sugary food and drinks at social gatherings, though wide confidence intervals are noted.

Given the observed intervention effect on diastolic blood pressure, we conducted post-hoc analyses of blood pressure measures. Namely, we evaluated intervention impact on pulse pressure, calculated as the difference between systolic and diastolic blood pressure, and mean arterial pressure, which accounts for flow, resistance and pressure in arteries and is calculated as diastolic blood pressure plus one third of the difference between diastolic and systolic blood pressure. Analysis showed a significant reduction in pulse pressure in intervention clusters relative to control (adjusted difference -1.97 (-3.77, -0.16), p = 0.0358). There was also a reduction in mean arterial pressure, though not statistically significant (-1.27 (-3.57, 1.02), p = 0.2403). Considering the demonstrated association between anxiety and hypertension in other contexts [29], and in recognition of the hypothesised effects of PLA via mechanisms of enhanced social support [30] we further analysed the GAD-7 measure of anxiety as a log-transformed continuous outcome as opposed to the categorical outcome included in our pre-specified analysis. This showed that there was a significant reduction in transformed GAD-7 score (-0.22 (-0.41, -0.04), p = 0.0201) in intervention clusters compared to control arm at endline. Data from post-hoc analyses are presented in S5 Table.

There was no evidence of interaction between the intervention and gender, household wealth, age, village size, or inclusion in baseline survey in relation to the primary outcome (S6 Table). Some potential interactions were identified, especially in relation to wealth and outcomes related to diabetes awareness, control, and complications, as well as anxiety and depression. Results suggest intervention effect on incidence of diabetes in the cohort of individuals with intermediate hyperglycaemia at baseline may be modified by wealth index, with poorer individuals benefiting more.

Total and average annual costs of the intervention were US$318,706 and US$157,696, respectively. The total cost per target population (adults ≥30 years) covered was US$12. Staff costs constituted 68% of the total intervention cost,

**Table 3. Frequencies and proportions, or means/medians, and absolute (coefficient) effects and 95% confidence interval comparing primary and secondary outcome measures between trial arms (intervention – control).**

| OUTCOMES | | Baseline | | Endline | | Crude Difference, mean or % (95%CI)* | Adjusted Difference, mean or % (95%CI)** |
|---|---|---|---|---|---|---|---|
| | | Control | Intervention | Control | Intervention | | |
| **PRIMARY** | | | | | | | |
| **Combined prevalence of intermediate hyperglycaemia and diabetes** | | 307 (45.1%) | 269 (38.2%) | 329 (49.9%) | 311 (45.5%) | -1.85 (-11.63, 7.92,); p=0.6777 | -1.19 (-10.74, 8.36,); p=0.7838 |
| **SECONDARY** | | | | | | | |
| **Blood pressure** | **Systolic blood pressure (mmHg), mean (sd)** | 133.5 (22.6) | 133.1 (22.1) | 127.0 (20.7) | 126.8 (19.1) | 0.02 (-3.20, 3.24); p=0.9900 | -0.15 (-3.33, 3.04); p=0.9176 |
| | **Diastolic blood pressure (mmHg), mean (sd)** | 75.7 (11.4) | 79.2 (11.4) | 75.8 (11.1) | 77.6 (10.8) | -2.78 (-5.24, -0.32,); p=0.0308 | -2.98 (-5.55, -0.41); p=0.0277 |
| | **Hypertension, n (%)** | 274 (39.9%) | 274 (38.8%) | 236 (35.8%) | 196 (28.7%) | -6.47 (-14.87, 1,93,); p=0.1153 | -7.40 (-16.03, 1.24,); p=0.0845 |
| **Overweight & obesity** | **Body Mass Index (BMI), mean (sd)** | 23.4 (4.0) | 22.7 (3.9) | 23.0 (4.0) | 22.8 (3.7) | 0.38 (-0.30, 1.07); p=0.2404 | 0.39 (-0.39, 1.17); p=0.2837 |
| | **Overweight or obese, n (%)** | 346 (50.4%) | 319 (45.3%) | 318 (48.2%) | 304 (44.5%) | 2.59 (-4.84, 10.0), p=0.4504 | 3.04 (-4.75, 10.83); p=0.4008 |
| | **Waist:Hip ratio, mean (sd)** | 0.90 (0.06) | 0.89 (0.07) | 0.91 (0.05) | 0.90 (0.07) | -0.01 (-0.04, 0.02); p=0.4133 | -0.01 (-0.04, 0.02); p=0.4420 |
| | **Abdominal obesity, n (%)** | 471 (68.7%) | 425 (60.2%) | 529 (80.2%) | 436 (63.8%) | -15.29 (-33.94, 3.36); p=0.0966 | -15.23 (-33.02, 2.56); p=0.0847 |
| **Dietary diversity score, mean (SD)** | | 5.4 (2.2) | 5.0 (1.5) | 4.7 (1.6) | 4.7 (1.4) | -0.10 (-0.59, 0.39); p=0.6566 | -0.09 (-0.59, 0.41); p=0.7000 |
| **Log transformed minutes spent engaged in physical activity per week, mean (sd)** | | 6.4 (0.8) | 6.3 (1.0) | 6.2 (0.7) | 5.9 (0.9) | -0.26 (-0.53, 0.01); p=0.0569 | -0.25 (-0.54, 0.04); p=0.0871 |
| **Diabetes knowledge** | **Ability to report one or more valid *causes* of diabetes (%)** | 346 (50.4%) | 291 (41.2%) | 336 (50.9%) | 576 (84.3%) | 33.73 (17.67, 49.79); p=0.0010 | 34.07 (18.05, 50.10); p=0.0010 |
| | **Ability to report one or more valid *symptoms* of diabetes (%)** | 417 (60.8%) | 321 (45.5%) | 374 (56.7%) | 595 (87.1%) | 35.4 (18.21, 52.59); p=0.0012 | 35.82 (18.67, 52.98); p=0.0011 |
| | **Ability to report one or more valid *complications* of diabetes (%)** | 213 (31.1%) | 225 (31.9%) | 196 (29.7%) | 534 (78.2%) | 48.17 (28.70, 67.64); p=0.0003 | 48.66 (29.32, 68.00); p=0.0003 |
| | **Ability to report one or more valid ways to *prevent* diabetes (%)** | 407 (59.3%) | 385 (54.5%) | 451 (68.3%) | 601 (88.0%) | 21.26 (10.51, 32.01); p=0.0015 | 21.82 (11.06, 32.58); p=0.0013 |
| | **Ability to report one or more valid ways to *control* diabetes (%)** | 453 (66.0%) | 410 (58.1%) | 498 (75.5%) | 606 (88.7%) | 14.75 (3.64, 25.86); p=0.0149 | 15.27 (4.16, 26.39); p=0.0125 |
| **PHQ-9 score ≥10 (moderate or severe depression), n (%)** | | 51 (7.4%) | 33 (4.7%) | 27 (4.1%) | 24 (3.5%) | -0.12 (-2.93, 2.69); p=0.9232 | -0.40 (-3.22, 2.42); p=0.7553 |
| **GAD-7 score ≥10 (moderate or severe anxiety), n (%)** | | 29 (4.2%) | 31 (4.4%) | 28 (4.2%) | 18 (2.6%) | -1.57 (-4.63, 1.49); p=0.2762 | -1.72 (-4.83, 1.38); p=0.2409 |
| **Self-awareness of diabetic status among individuals identified with diabetes by blood glucose measures, n (%)** | | 70 (49.3%) | 47 (43.5%) | 69 (47.9%) | 51 (48.1%) | 0.77 (-20.28, 21.81); p=0.9361 | 3.61 (-14.89, 22.10); p=0.6694 |
| **Utilisation of services for treatment or advice for diabetes among individuals with prior diagnosis of diabetes** | | 63 (90.0%) | 40 (85.1%) | 66 (95.7%) | 46 (90.2%) | -4.33 (-15.66, 6.99); p=0.4093 | -3.92 (-14.63, 6.79); p=0.4288 |
| **Two-year diabetes incidence among individuals with intermediate hyperglycaemia at baseline, n (%)** | | | | 25 (17.1%) | 12 (9.5%) | -8.60 (-20.91, 3.72); p=0.1509 | -8.60 (-20.09, 2.89); p=0.1265 |

\* Adjustment for baseline outcome measure at the cluster level. **Adjusted for cluster-level baseline outcome measure, gender and age as linear and quadratic terms. Intraclass correlation (ICC) of primary outcome at endline is 0.01.

**Table 4. Frequencies, proportions, means/medians and absolute (coefficient) effects and 95% confidence interval comparing explanatory outcome measures by trial arm (intervention – control).**

| OUTCOMES | | Baseline | | Endline | | Crude Difference (95%CI)* | Adjusted Difference (95%CI)** |
|---|---|---|---|---|---|---|---|
| | | Control | Intervention | Control | Intervention | | |
| **Self-rated health, mean (sd)** | | 76.1 (17.6) | 86.4 (12.8) | 69.2 (16.6) | 78.0 (18.4) | 8.33 (1.91, 14.75); p=0.0167 | 9.12 (2.58, 15.6); p=0.0116 |
| **Daily smoking or use of smokeless tobacco (%)** | | 218 (31.8%) | 275 (39.0%) | 242 (36.7%) | 259 (37.9%) | -2.26 (-10.87, 6.35); p=0.5670 | -1.44 (-11.1, 8.23); p=0.7439 |
| **Betel nut use (%)** | | 207 (30.2%) | 261 (37.0%) | 212 (32.1%) | 235 (34.4%) | 1.26 (-11.96, 14.5); p=0.8344 | 0.25 (-12.61, 13.1); p=0.9657 |
| **Log sedentary time in previous 24 hours, median (IQR)** | | 4.7 (4.4-5.0) | 4.7 (4.1-5.2) | 4.7 (4.1-5.1) | 5.0 (4.8-5.2) | 0.26 (0.05, 0.47); p=0.0206 | 0.24 (0.0, 0.43); p=0.0225 |
| **Log time engaged in brisk walking, median (IQR)** | | 5.3 (4.8-6.0) | 5.1 (4.8-5.6) | 5.3 (4.8-5.5) | 5.2 (4.8-5.5) | -0.15 (-0.52, 0.22); p=0.3777 | -0.13 (-0.49, 0.24); p=0.4454 |
| **Among individuals with prior diagnosis of diabetes** | **Diabetes control (%)** | 15 (21.4%) | 7 (14.9%) | 15 (21.7%) | 15 (29.4%) | 3.56 (-12.16, 19.28); p=0.6208 | 1.80 (-15.00, 18.60); p=0.8139 |
| | **Diabetes affect (ADS) score, mean (sd)** | 14.9 (3.3) | 13.8 (3.2) | 16.1 (3.8) | 16.5 (3.5) | 0.90 (-1.47, 3.26); p=0.4134 | 0.78 (-1.58, 3.14); p=0.4716 |
| | **Diabetes-related complications (%)** | 55 (78.6%) | 28 (59.6%) | 60 (87.0%) | 33 (64.7%) | -22.54 (-48.23, 3.15); p=0.0784 | -22.46 (-48.22, 3.30); p=0.0801 |
| | **At least monthly blood glucose testing (%)^** | 17 (24.3%) | 17 (36.2%) | 18 (26.1%) | 17 (33.3%) | 25.06 (-23.55, 73.68); p=0.2735 | 19.97 (-29.74, 69.68); p=0.3871 |
| | **Stigma 1: Perceived inability to fulfil responsibilities+** | 31 (44.3%) | 12 (25.5%) | 32 (46.4%) | 25 (49.0%) | 22.90 (-18.29, 64.09); p=0.2402 | 23.18, (-17.25, 63.62); p=0.2269 |
| | **Stigma 2: Perceived to be seen as a lesser person+** | 18 (25.7%) | 4 (8.5%) | 13 (18.8%) | 14 (27.5%) | 18.41 (-35.51, 72.32); p=0.4598 | 18.08 (-32.23, 68.39); p=0.4373 |
| | **Stigma 3: Embarrassed in social situations+** | 29 (41.4%) | 5 (10.6%) | 20 (29.0%) | 17 (33.3%) | 38.40 (-8.21, 85.01); p=0.0953 | 37.17 (-8.32, 82.65); p=0.0976 |
| | **Stigma 4: Ashamed of having diabetes+** | 18 (25.7%) | 4 (8.5%) | 11 (15.9%) | 9 (17.7%) | 8.79 (-16.79, 34.38); p=0.4567 | 10.42 (-17.29, 38.13); p=0.4171 |
| | **Feels supported by family in managing diabetes~** | 58 (82.9%) | 28 (59.6%) | 49 (71.0%) | 36 (70.6%) | -5.64 (-27.55, 16.27); p=0.5747 | -5.05 (-27.15, 17.04); p=0.6173 |
| | **Ever experienced physical, social, or mental abuse due to diabetic status** | 27 (38.6%) | 5 (10.6%) | 20 (29.0%) | 17 (33.3%) | 7.45 (-31.09, 45.99); p=0.6723 | 6.81 (-32.38, 46.00); p=0.7035 |
| **Social norms: Perceived negative feelings or family or social reactions to:** | **1) Going on a morning walk alone (women only)** | 35 (8.0%) | 87 (19.9%) | 26 (6.2%) | 105 (25.1%) | 4.47 (-5.12, 14.06); p=0.3192 | 4.19 (-5.77, 14.1); p=0.3662 |
| | **2) Going for a morning walk with a female relative (women only)** | 10 (2.3%) | 26 (5.9%) | 15 (3.6%) | 74 (17.7%) | 7.87 (-2.43, 25.2); p=0.3302 | 8.12 (-8.90, 25.15); p=0.3084 |
| | **3) Eating less than usual or refusing oily or sugary foods and drinks at social gatherings** | 494 (72.0%) | 580 (82.2%) | 462 (70.0%) | 612 (89.6%) | 14.36 (2.40, 26.32); p=0.0237 | 14.59 (2.25, 26.93); p=0.0254 |
| | **4) Providing healthy snacks to guests** | 392 (57.1%) | 305 (43.2%) | 356 (53.9%) | 392 (57.4%) | 4.02 (-28.58, 36.62); p=0.7866 | 4.07 (-28.49, 36.63); p=0.7839 |

* Adjustment for baseline outcome measure at the cluster level. **Adjusted for cluster-level baseline outcome measure, gender and age as linear and quadratic terms. ^Missing data for 9 cases – 1 in control arm at baseline, 4 in each arm at endline. +Missing data for 7 cases – 3 in intervention arm and 1 in control arm at baseline, 2 in intervention arm and 1 in control arm at endline ~Reports at least moderate family support

followed by other recurrent costs (e.g., overheads) and COVID-19 protection measures, accounting for 13% and 11%, respectively. Costs of the activities conducted in the intervention start-up period (16 months) accounted for around 12% of total costs. S7 Table provides breakdown of the intervention costs by inputs and activities. ICER was US$4.2

per 1 mmHg reduction in diastolic blood pressure, ranging from \$2.2 to \$30.3 using the confidence interval around the effect size.

## Discussion

We evaluated the effects of facilitated PLA community mobilisation on T2DM and non-communicable disease risk among a general population when horizontally scaled-up and in a dynamic context of COVID-19 in rural Bangladesh. There is no evidence of intervention effect on the primary outcome of prevalence of T2DM and intermediate hyperglycaemia. The intervention is associated with large improvements in diabetes knowledge as well as changes in diastolic blood pressure and self-rated health. Potentially important reductions in the prevalence of hypertension and abdominal obesity, as well as self-reported diabetes-related health complications are also noted, though these do not reach statistical significance. Among a higher-risk population cohort (i.e., individuals with intermediate hyperglycaemia at baseline), the direction and magnitude of intervention effect on incidence of diabetes among is very similar to that observed in our former DMagic trial (-8.7 (-3.5, -14.0)) [6], though given our smaller sample size in D:Clare, the wider confidence intervals included the null.

The absence of a clear intervention effect on the prevalence of intermediate hyperglycaemia and T2DM differs from what we observed in the DMagic trial, which showed large significant effects [6]. However, the intervention also differed in important ways as a result of COVID-19 disruptions and adaptations. Our process evaluation allows us to examine what worked well and, crucially, to analyse fidelity to the PLA intervention that we planned, and factors affecting this.

There were three main ways that the intervention was not implemented as planned because of COVID-19 disruptions and delays: 1. Phases of the intervention were shorter; 2. Group size, average attendance at PLA groups, interaction between group members and facilitators, and between group members was restricted; 3. Community action was not initiated on a large scale. The lack of pre-existing social cohesion in our study context, which was exacerbated by COVID-19 restrictions, has been reported in other studies as a factor which makes it challenging to mobilise communities for social change [31,32]. The shorter time frame for the first phase of the intervention and interruptions to meeting schedules may also have prevented the development of a core group of members to drive action. A lack of community cohesion in intervention areas fed into a lack of group cohesion, and when combined with restrictions on group size and less consistency in group membership, it was difficult to develop collective responsibility for prevention and control of diabetes. Our intervention was conducive to participatory learning and awareness raising as reflected in diabetes-related awareness outcomes but may not have allowed sufficient time for collective action which usually occurs after the wider community meeting. Compared to DMagic, where 74% of survey respondents in intervention areas reported ever participating in PLA meetings, repeat attendance was high, and 72% knew someone who participated [6], participation in and wider awareness of the intervention was low in D:Clare. Details and results from an in-depth mixed-methods process evaluation of the D:Clare intervention are presented elsewhere [22].

We have previously described one of the key mechanisms of PLA as a process of enabling individuals to act within enabled, supportive environments [30,7]. It may be that without adequate time devoted to development of community action and implementation of locally agreed strategies our D:Clare implementation was unable to translate raised awareness into effective collective actions. The findings of increased population awareness without impact on T2DM outcomes is similar to the effect of the mHealth health promotion intervention component of DMagic, which similarly lacked opportunity for collective action [33]. Nevertheless, knowledge and awareness are well-established components of health literacy [34]. This, in turn, supports individuals and communities in preventing and controlling disease, and is necessary to accelerate progress towards global non-communicable disease targets [34,35].

We observed a statistically significant reduction in diastolic blood pressure and a suggested reduction in hypertension prevalence in D:Clare, though we note some caution in our certainty of effect due to the number of outcomes

tested. Concurrent PLA impact on abdominal obesity and log-transformed anxiety score may have been important mechanisms for this though further exploration of this and the possible role of salt consumption and physical activity changes are needed. While no effects on blood pressure outcomes were observed in the DMagic trial, we did observe similar impacts in our five-year post-randomisation follow-up of DMagic communities [9]. Raised blood pressure and hypertension is the leading modifiable risk factor for future morbidity and mortality from cardiovascular diseases, specially heart attacks and stroke [36,37] and blood pressure lowering provides broadly generalisable benefits [38]. Diastolic blood pressure, pulse pressure and mean arterial pressure are related to peripheral vascular resistance and loss of arterial vasodilatory mechanisms and have been associated with ischaemic stroke [39–41]. Data from the Framingham study in the 1990s suggest that even modest reductions in mean population diastolic blood pressure could result in substantial reductions in the prevalence of hypertension and associated risks of cardiovascular events [42]. The potential to achieve such reductions in blood pressure through population-based interventions of health messaging and community-based strategies such as PLA warrants further exploration, especially in settings that lack screening and treatment for raised blood pressure.

The large, rural, population-based surveys with high response rates at baseline and endline, and the high follow-up of individuals identified with intermediate hyperglycaemia at baseline, with assessment of glycaemic status through fasting and two-hour blood glucose tests are notable strengths of our trial. The inability to ensure blinding of enumerators and participants in the endline survey is a potential weakness. Not all of the same enumerators worked on the endline and baseline surveys and enumerators worked within specific clusters, therefore some inter-observer variation is possible. The use of self-reported measures of behaviours is also a potential weakness of our design and may be subject to recall and social desirability bias and it is plausible that this would vary by trial arm. Measures of physical activity, dietary practice, care-seeking and attitudes must therefore be interpreted with caution. Finally, there was overlap between the timing of baseline data collection and the start of intervention implementation, however, it unlikely that the PLA intervention would have any impacts in this early stage.

The D:Clare intervention cost $12 per target population (adults ≥30 years). This is slightly higher than DMagic intervention, mainly due to longer implementation period (20 vs 18 months) and cost of COVID-19 protection measures. At US$4.2 per 1 mmHg reduction in diastolic blood pressure, ranging from $2.2 to $30.3, the ICER for D:Clare is significantly lower than previous community-based hypertension control interventions [43].

## Conclusion

Scale-up of PLA for T2DM prevention and control during the evolving COVID-19 pandemic required adaptations to intervention implementation and evaluation approach. There was no evidence of impact of the adapted intervention on blood glucose outcomes, but large and important impacts on diabetes knowledges and likely effects on other health outcomes, including blood pressure. Process evaluation combined with cluster RCT data provide important insights into intervention and implementation features which are necessary for the success of the intervention. Critical to this is providing sufficient time for intervention delivery that allows communities to develop and share collective knowledge and plan and implement collective action. The PLA approach to non-communicable disease prevention and control continues to show promise.

## Supporting information

**S1 Table. Participatory Learning and Action (PLA) group design and adaptations in response to COVID-19.** (DOCX)

**S2 Table. Glycaemic definitions and diagnostic criteria used in the D:Clare trial, adapted from WHO 2006.** (DOCX)

**S3 Table. D:Clare trial secondary outcome measures and definitions.**
(DOCX)

**S4 Table. Sensitivity analysis of the primary outcome showing frequencies, proportions, estimated difference (intervention minus control) and 95% confidence interval comparing prevalence of intermediate hyperglycaemia and diabetes between trial arms using different outcome definitions.**
(DOCX)

**S5 Table. Post-hoc analysis of intervention effects on pulse pressure, mean arterial pressure and log-transformed anxiety (GAD-7) score.**
(DOCX)

**S6 Table. Interaction odds ratios and regression coefficients (95% CIs) between trial arm and gender, wealth index, age, village size and inclusion in baseline sample for primary and secondary study outcomes.** Odds ratios (for binary outcomes) and coefficients (for continuous outcomes) indicate the difference in intervention effect on outcomes between interaction categories.
(DOCX)

**S7 Table. Intervention costs disaggregated by input, activities and implementation phase.**
(DOCX)

**S1 Checklist. CONSORT 2010 checklist of information to include when reporting a cluster randomised trial.**
(DOCX)

**S2 Checklist. Inclusivity in global research questionnaire.**
(DOCX)

**S1 Text. Statistical analysis plan.**
(PDF)

## Acknowledgments

The study team would like to thank the D:Clare Trial Steering Committee (David Beran (Chair), Graham Hitman, Justine Davies, Edward Gregg, Aaron Holiday, Jennifer Thompson, Calum Davey, Audrey Prost and Anthony Costello) and the Data Monitoring Committee (Mohammod Shahidullah (Chair), Meena Daivadanam, and A.T.M Iqbal Anwar) for their valuable contributions to the project. We are grateful to our Community Advisory Groups and Md. Golam Azam for contributions to field activities.

## Author contributions

**Conceptualization:** Edward Fottrell, Joanna Morrison, Carina King, Hassan Haghparast-Bidgoli, A.K. Azad Khan, Kishwar Azad.

**Data curation:** Joanna Morrison, Carina King, Sanjit Kumer Shaha MSS, James Beard.

**Formal analysis:** Edward Fottrell, Joanna Morrison, Carina King, Malini Pires, Sarker Ashraf Uddin Ahmed, Andrew Copas, Hassan Haghparast-Bidgoli.

**Funding acquisition:** Edward Fottrell, Abdul Kuddus, Joanna Morrison, Carina King, Andrew Copas, Hassan Haghparast-Bidgoli, A.K. Azad Khan, Kishwar Azad.

**Investigation:** Joanna Morrison, Tasmin Nahar MSS, Carina King, Sarker Ashraf Uddin Ahmed, Naveed Ahmed, Andrew Copas, Hassan Haghparast-Bidgoli.

**Methodology:** Edward Fottrell, Tasmin Nahar MSS, Carina King, Sanjit Kumer Shaha MSS, Malini Pires, Sarker Ashraf Uddin Ahmed, Naveed Ahmed, Andrew Copas, Hassan Haghparast-Bidgoli.

**Project administration:** Edward Fottrell, Abdul Kuddus, Kishwar Azad.

**Supervision:** Edward Fottrell, Abdul Kuddus, Joanna Morrison, Tasmin Nahar MSS, Carina King, Sanjit Kumer Shaha MSS, Sarker Ashraf Uddin Ahmed, James Beard, Naveed Ahmed, Hassan Haghparast-Bidgoli, A.K. Azad Khan, Kishwar Azad.

**Validation:** Carina King.

**Writing – original draft:** Edward Fottrell.

**Writing – review & editing:** Edward Fottrell, Abdul Kuddus, Joanna Morrison, Tasmin Nahar MSS, Sanjit Kumer Shaha MSS, Malini Pires, Sarker Ashraf Uddin Ahmed, James Beard, Naveed Ahmed, Andrew Copas, Hassan Haghparast-Bidgoli, A.K. Azad Khan.

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
