## [Decision Letter · Decision Letter 0]

26 Feb 2025

PGPH-D-24-01381

A cluster randomised controlled trial of community groups using Participatory Learning and Action to prevent and control diabetes and intermediate hyperglycaemia in rural Bangladesh.

Dear Dr. Fottrell,

Thank you for submitting your manuscript to PLOS Global Public Health. After careful consideration, we feel that it has merit but does not fully meet PLOS Global Public Health’s publication criteria as it currently stands. Therefore, we invite you to submit a revised version of the manuscript that addresses the points raised during the review process.

We look forward to receiving your revised manuscript.

Kind regards,

Abdur Razzaque Sarker, PhD

Academic Editor

Journal Requirements:

2.  In the online submission form, you indicated that "Deidentified data collected for this study and a data dictionary are available from the corresponding author on reasonable request.". 

3. Uploaded as supplementary information.

Additional Editor Comments (if provided):

The authors have presented a compelling topic with relevant data, but there are critical issues that need to be addressed. These include some revisions in the introduction, some clarifications in the method, result and conclusion sections. Additionally, some sections of the manuscript require better clarity to improve readability and coherence.

Reviewers' comments:

Reviewer's Responses to Questions

**Comments to the Author**

1. Does this manuscript meet PLOS Global Public Health’s publication criteria?

Reviewer #1: Yes

Reviewer #2: Yes

Reviewer #3: Partly

2. Has the statistical analysis been performed appropriately and rigorously?

Reviewer #1: Yes

Reviewer #2: Yes

Reviewer #3: Yes

3. Have the authors made all data underlying the findings in their manuscript fully available (please refer to the Data Availability Statement at the start of the manuscript PDF file)?

Reviewer #1: Yes

Reviewer #2: No

Reviewer #3: Yes

4. Is the manuscript presented in an intelligible fashion and written in standard English?

Reviewer #1: Yes

Reviewer #2: Yes

Reviewer #3: Yes

Reviewer #1: Major comments

Introduction

The authors are suggested to include global and national data related to the title and provide a rationale for the study, which is currently missing from the background.

Method

In the randomisation and masking subheading, the authors stated that ‘Randomisation took place at a project orientation meeting in Faridpur town attended by community representatives and independent observers.’

The authors are suggested to clarify on that whether the community representatives and independent observers were involved in the project at a later stage.

In the piloting & interruption subheading, ‘Experiences from the pilot were shared with study Community Advisory Groups – groups based in each of the study upazilas and comprised of community stakeholders – who were asked to provide input on the acceptability of implementing the PLA intervention after COVID-19 lockdown.’

Did the authors revised the PLA intervention based on their feedbacks? Please mention that.

In the individual questionnaire subheading, the authors stated that they used a mental health screening tool, but they did not mention their rationale for using it earlier in the manuscript. Was the tool validated from the perspective of Bangladesh? If not, how did they validate it in the context of Bangladesh? Please provide clarification.

In the physical measurement subheading, the authors mentioned Measurements of height, weight, and waist and hip girth were taken with light clothes without shoes and using standard methods. Need to cite a reference there.

In the sample size subheading, the authors mentioned that (4) village size (smaller vs. larger villages). How did they categorize it? For example: 4 or fewer villages considered a small and more than that considered large villages?

Result

From Line 377, page no. 9, the authors showed some data but it’s not clear which table represents the data as they described Table 4 from their previous sentence. The data shown is from Table 3. Please check it.

On page 14, starting from Line 405, the authors discussed post-hoc analyses of blood pressure and presented the results. However, they did not refer to any supplementary table or provide one. Therefore, it is recommended that the authors add a supplementary file with the relevant blood pressure data.

From Line 432, page no. 14, the authors provided breakdown of the intervention costs though the supplementary file 4 showed the sensitivity analysis data.

From Line 432, page no. 14, the authors detailed the intervention costs, while the supplementary file 4 showed the sensitivity analysis data. Please check it. However, Supplementary Table 6 displayed intervention costs data. Please verify and make the necessary corrections.

Discussion

The authors referred to Table 2 in the discussion section, but it is recommended not to include result tables in the discussion section.

From Lines 477-481, on page 15, the authors are advised to break it into two sentences, which will make it easier to understand.

Conclusion

The authors are suggested to include recommnedation on how to make their project sustainable and how to integrate it into the health policy system.

Minor comments

Abstract

Participants

The authors mentioned Adults aged >30 years who permanently reside in study villages, some places they mentioned �30 years. Please check this all over the manuscript and correct it.

Intervention

The authors started with 108 PLA groups were created. The authors are suggested to splled it out. Example: One hundred and eight.

Method section of the main manuscript

The authors may include the study duration if the word count permits.

In the main manuscript, in the randomisation and masking subheading, the authors mentioned 12 folded pieces of paper. Please spell out 12 with Twelve.

In the main manuscript, in the intervention subheading, the authors mentioned 108 PLA groups, 9 men’s and 9 women’s in each intervention cluster, please spell out 108 with one hundred and eight.

In sattistical analysis subheading, please add the Stata version.

Reviewer #2: Many thanks for a clear, well-written article.

Where are the data able to be viewed?

Line 512: please re-read this line as I think it has an extra “covered was $12”.

It was great to be able to work there.

Reviewer #3: The manuscript is written well. The background section is missing and should be included. On the ethical concerns please indicate what happened to individuals who had higher blood glucose and blood pressure that needs treatment, at both clusters in baseline and endline. If possible strengthen the discussion with other similar studies beside the DMagic study output. And explore what could other possible reasons in addition to the COVID-19 adaptations have an effect in the outcome measures. Support some of the results with qualitative findings if available at hand.

**Do you want your identity to be public for this peer review?** For information about this choice, including consent withdrawal, please see our Privacy Policy

Reviewer #1: **Yes: ** Yasmin Jahan

Reviewer #2: No

Reviewer #3: **Yes: ** Gadise Bekele Regassa

---

## [Decision Letter · Decision Letter 1]

13 Jul 2025

PGPH-D-24-01381R1

A cluster randomised controlled trial of community groups using Participatory Learning and Action to prevent and control diabetes and intermediate hyperglycaemia in rural Bangladesh.

Dear Dr. Fottrell,

Thank you for submitting your manuscript to PLOS Global Public Health. After careful consideration, we feel that it has merit but does not fully meet PLOS Global Public Health’s publication criteria as it currently stands. Therefore, we invite you to submit a revised version of the manuscript that addresses the points raised during the review process.

There are a few more minor revision comments to address from Reviewer 5. Can you please take a look and respond as best you can? Thank you for your patience through the review process.

We look forward to receiving your revised manuscript.

Kind regards,

Julia Robinson

Executive Editor

Journal Requirements:

3. Please provide separate main figure files in .tif or .eps format only and remove any figures embedded in your manuscript file. Please also ensure that all files are under our size limit of 10MB. Please leave the figure captions or legends in the manuscript.

4. We notice that your supplementary tables are included in the manuscript file. Please remove them and upload them with the file type 'Supporting Information'. Please ensure that each Supporting Information file has a legend listed in the manuscript before or after the references list.

5. We have noticed that you have uploaded Supporting Information files, but you have not included a list of legends. Please add a full list of legends for your Supporting Information files before or after the references list.

6. Please amend your online detailed Financial Disclosure statement. This is published with the article. It must therefore be completed in full sentences and contain the exact wording you wish to be published.

a) State the initials, alongside each funding source, of each author to receive each grant, if applicable. For example: "This work was supported by the National Institutes of Health (####### to AM; ###### to CJ) and the National Science Foundation (###### to AM)."

7. Please ensure that the funders and grant numbers match between the Financial Disclosure field and the Funding Information tab in your submission form. Note that the funders must be provided in the same order in both places as well.

Additional Editor Comments (if provided):

Reviewers' comments:

Reviewer's Responses to Questions

**Comments to the Author**

Reviewer #2: All comments have been addressed

Reviewer #4: All comments have been addressed

Reviewer #5: All comments have been addressed

publication criteria?

Reviewer #2: (No Response)

Reviewer #4: Yes

Reviewer #5: Yes

3. Has the statistical analysis been performed appropriately and rigorously?

Reviewer #2: (No Response)

Reviewer #4: Yes

Reviewer #5: Yes

4. Have the authors made all data underlying the findings in their manuscript fully available (please refer to the Data Availability Statement at the start of the manuscript PDF file)?

Reviewer #2: (No Response)

Reviewer #4: Yes

Reviewer #5: Yes

5. Is the manuscript presented in an intelligible fashion and written in standard English?

Reviewer #2: (No Response)

Reviewer #4: (No Response)

Reviewer #5: No

Reviewer #2: (No Response)

Reviewer #4: The revised manuscript is accepted in its present form.

Reviewer #5: R1:

• This study is unique and depend on details but those information should be presented professionally.

• It is important to highlight and know the importance of awareness

• Over all the topic there was elaboration and more undesired details.

• I face some difficulties in understanding the idea of the topic.

• Presentation should be simple, clear and easy to understand.

Title:

• summarize the title. suggestion of short title: (effect of participatory learning and action in regulation of blood glucose)

• using of abbreviation in title is not preferable and ambiguous.

Abstract:

• Covid 19- is one of the two important study objectives?

• key word must be novel or infrequent.

• suggestion: diabetes, community intervention

materials and methods:

1. write in specific titles with brief and meaning ful manner.

2. rearrange methods as to give complete idea in each paragraph regarding Study design and area, Study subjects, Inclusion criteria, Exclusion criteria, methods for data Collection and Preparation of Samples and grouping of study participants.

3. In each sub title author must write all date regarding the specific paragraph for one time to avoid repetition of information and to divide the research paper in well known title(abstract, introduction, methods….etc)

4. set your participants in methodology as groups i.e.; diabetic group, , control group, intervention group, other non communicable disease group

5. write how the author group the study participant.

6. ignore any participant who were not in the inclusion criteria. Write the exact number of study participants only.

7. Give explanation for covid 19 include its impact in study participant

8. If the covid 19 effect the sample size

9. mention the noncommunicable disease included in current study.

Results:

• write every result of specific table above the table then jump to the next table.

Discussion:

Good, but there is elaboration in this section.

**Do you want your identity to be public for this peer review?** For information about this choice, including consent withdrawal, please see our Privacy Policy

Reviewer #2: No

Reviewer #4: No

Reviewer #5: **Yes: ** Nahla Ahmed Mohammed Abdelrahman

---

## [Editor Report · Decision Letter 2]

28 Jul 2025

A cluster randomised controlled trial of community groups using Participatory Learning and Action to prevent and control diabetes and intermediate hyperglycaemia in rural Bangladesh.

PGPH-D-24-01381R2

Dear Professor Fottrell,

We are pleased to inform you that your manuscript 'A cluster randomised controlled trial of community groups using Participatory Learning and Action to prevent and control diabetes and intermediate hyperglycaemia in rural Bangladesh.' has been provisionally accepted for publication in PLOS Global Public Health.

Best regards,

Julia Robinson

Executive Editor